# Effects of Exogenous Melatonin on Root Physiology, Transcriptome and Metabolome of Cotton Seedlings under Salt Stress

**DOI:** 10.3390/ijms23169456

**Published:** 2022-08-21

**Authors:** Wenjing Duan, Bin Lu, Liantao Liu, Yanjun Meng, Xinying Ma, Jin Li, Ke Zhang, Hongchun Sun, Yongjiang Zhang, Hezhong Dong, Zhiying Bai, Cundong Li

**Affiliations:** 1State Key Laboratory of North China Crop Improvement and Regulation, College of Life Science, Hebei Agricultural University, Baoding 071000, China; 2State Key Laboratory of North China Crop Improvement and Regulation, Key Laboratory of Crop Growth Regulation of Hebei Province, College of Agronomy, Hebei Agricultural University, Baoding 071000, China; 3College of Landscape and Tourism, Hebei Agricultural University, Baoding 071000, China; 4Cotton Research Center, Key Laboratory of Cotton Breeding and Cultivation in Huang-Huai-Hai Plain, Ministry of Agriculture, Shandong Academy of Agricultural Sciences, Jinan 250100, China

**Keywords:** melatonin, salt stress, cotton root system, transcriptome, metabolomics

## Abstract

Root systems are the key organs through which plants absorb water and nutrients and perceive the soil environment and thus are easily damaged by salt stress. Melatonin can alleviate stress-induced damage to roots. The present study investigated the effects of exogenous melatonin on the root physiology, transcriptome and metabolome of cotton seedlings under salt stress. Salt stress was observed to damage the cell structure and disorder the physiological system of cotton seedling roots. After subjecting melatonin-soaked seeds to salt stress, the activities of SOD, CAT and POD in cotton seedling roots increased by 10–25%, 50–60% and 50–60%, respectively. The accumulation of H_2_O_2_ and MDA were significantly decreased by 30–60% and 30–50%, respectively. The contents of soluble sugar, soluble protein and K^+^ increased by 15–30%, 15–30% and 20–50%, respectively, while the Na^+^ content was significantly reduced. Melatonin also increased auxin (by 20–40%), brassinosteroids (by 5–40%) and gibberellin (by 5–35%) and promoted melatonin content and root activity. Exogenous melatonin maintained the integrity of root cells and increased the number of organelles. Transcriptomic and metabolomic results showed that exogenous melatonin could mitigate the salt-stress-induced inhibition of plant root development by regulating the reactive oxygen species scavenging system; ABC transporter synthesis; plant hormone signal transduction, endogenous melatonin gene expression; and the expression of the transcription factors MYB, TGA and WRKY33. These results provide a new direction and empirical basis for improving crop salt tolerance with melatonin.

## 1. Introduction

Salinized land exists in at least 75 countries worldwide, accounting for more than 20 percent of the world’s irrigated area, and this proportion has been increasing over time [1]. Salt stress inhibits crop growth, thus decreasing crop yields [2]. Roots play an important role in physically securing plants and absorbing water and nutrients. Abiotic stress on plants generally acts on different organs, among which roots are the first to be affected [3,4]. Under normal growth conditions, roots can absorb water and nutrients and transport them to the above-ground parts of plants to maintain the dynamic balance of cells, but under salt stress, the root phenotype and cell structure are changed, thus disrupting this balance [5,6]. Salt stress has been shown to inhibit the growth of plant roots, resulting in decreases in root length, root surface area and other related phenotypes, thus slowing root growth. At the same time, salt stress can reduce the absorption of nutrients by plant roots, weaken the water absorption capacity of plants and affect the water supply of roots to the above-ground parts, thus affecting the overall growth and development of plants [7]. Accordingly, salt stress results in a large number of changes in the structure and composition of plant cells, including increased vacuolation, nuclear chromatin deformation and plasma membrane damage [8]. Salt stress can lead to increases in the levels of reactive oxygen species (ROS), including superoxide anion, hydrogen peroxide, hydroxyl radical, ozone and singlet oxygen in roots, and the normal root metabolism is severely damaged by ROS, resulting in oxidative damage to root cells [9]. The auxin balance and distribution pattern in Arabidopsis taproots has been shown to be disturbed by salt stress, with the growth of taproots also inhibited [10]. Transcription factors in plant cells are also very important elements of plant responses to salt stress. Two types of transcription factors and genes have been identified to be involved in the salt stress response. One group is comprised of the transcription factors NAC, AP2, MYB and WRKY, which are mainly involved in salt stress responses and regulate root growth and development [11,12,13,14]. The other group is comprised of NHX and NHA, which regulate ion metabolism and transport, as well as peroxidase (POD) and catalase (CAT), which are related to the response to salt stress and ROS scavenging [15,16]. These proteins are all involved in the response of plant roots to salt stress [11,12,13,14,15,16].

Melatonin is an indole that occurs throughout plant species and participates in regulating plant physiological processes [17]. As a strong free radical scavenger [18], melatonin participates in antioxidant systems, promotes root development and alleviates damage induced by stress. Additionally, melatonin can control root morphology by regulating auxin expression [19]. The taproots and root hairs of Arabidopsis are suppressed by melatonin, and their lateral roots and adventitious roots are stimulated [20,21]. Melatonin can also activate the auxin signaling pathway and promote the formation and development of lateral roots in rice [22]. The activities of antioxidant enzymes (i.e., superoxide dismutase (SOD), POD and CAT) and soluble protein levels were significantly increased in the roots of strawberry seedlings, and the accumulation of ROS was significantly inhibited after melatonin treatment [23]. After melatonin treatment under salt stress, ROS metabolism and the expression of genes related to antioxidant defense were up-regulated, thus improving the salt resistance of rapeseed [24]. Melatonin and auxin have the same precursor substance, tryptophan, and there are four steps in the synthesis of melatonin from its tryptophan precursor, among which six genes, namely TDC, TPH, T5H, SNAT, ASMT and COMT, are involved in the synthesis of melatonin in plants [21]. There are two organelles in plants, mitochondria and chloroplasts, which are the main sites for the synthesis of plant melatonin [25]. The exogenous application of melatonin can induce the accumulation of endogenous melatonin content and enhance plant resistance [26]. Transcriptome enrichment analysis showed that melatonin regulates plant metabolic balance through nitrogen metabolism, hormone metabolism and the tricarboxylic acid cycle [27]. Exogenous melatonin can, by increasing the contents of amino acids and organic acids, including through the accumulation of metabolites, enhance plants’ resistance to abiotic stress [28]. Melatonin can also regulate the synthesis of secondary metabolites to regulate the senescence of plants, indicating that melatonin can regulate the balance between the primary and secondary metabolisms of plants, thus regulating their stress resistance [29].

Cotton is the most important natural textile fiber crop and plays an important role in agricultural production, industry and human life more broadly. It has some degree of salt tolerance and is grown in large quantities in saline–alkali soil conditions [30]. However, the seedling stage is the critical stage of development in which cotton is vulnerable to salt stress, and thus, salt can affect plant development and yield formation in cotton. We have previously studied the effects of exogenous melatonin on the seed germination and shoot growth of cotton under stress [26,31] and confirmed that exogenous melatonin promotes the activity of the root system of cotton seedlings under salt stress [32], but the regulatory mechanism by which exogenous melatonin affects the root system of cotton seedlings under salt stress has remained unclear. In this study, cotton seedlings were treated with 10 μmol∙L^−1^ melatonin and 150 mmol∙L^−1^ NaCl. Then, antioxidant enzymes, osmotic regulation substances and hormones in the roots of cotton seedlings were assayed. Furthermore, RNA-seq and metabolome techniques were used to analyze the differentially expressed genes (DEGs) and metabolites in cotton associated with salt stress under melatonin treatment. The purposes of this study were (1) to identify the antioxidant system and hormone content changes and root ultrastructure characteristics of the root system of melatonin-treated cotton seedlings that alleviated salt stress and (2) to identify the key root transcription factors and metabolites that alleviated salt stress, so as to clarify the internal mechanism by which melatonin regulates the root systems of cotton seedlings under salt stress. 

## 2. Results

### 2.1. Effects of Exogenous Melatonin on Cotton Seedlings and Their Root Morphology under Salt Stress

Under salt stress (S and MS), the shoots and roots of cotton seedlings grew slowly, and the plants were short with underdeveloped roots. Exogenous melatonin was able to promote the growth and development of the shoots and roots of MT and MS cotton seedlings, as shown in Figure 1. At 5, 10 and 15 d of treatment, the plant height and root morphology indices of the cotton seedlings under salt treatment were significantly reduced (Figure 1 and Figure 2). 

Cotton seedling root growth under salt stress (S and MS) was inhibited at 5, 10 and 15 d of treatment, and exogenous melatonin promoted root growth and development (MT and MS). On the 15th day, compared with the CK treatment, the total root volume, total lateral root number, average taproot diameter, taproot surface area and taproot volume under the MT treatment increased significantly by 37.84%, 19.40%, 38.03%, 81.55% and 11.32%, respectively; additionally, total root length, total surface area, total volume, total lateral root number and taproot volume were significantly decreased by 40.30%, 45.42%, 47.57%, 31.97% and 35.85% under the S treatment, respectively. Total root length, total root surface area, total root volume, total lateral root number, average taproot diameter, taproot surface area and taproot volume under the MS treatment were significantly increased by 56.94%, 49.56%, 75.26%, 35.54%, 18.88%, 57.99% and 32.35%, respectively, compared with the S treatment (*p* < 0.05). 

Thus, salt stress inhibited the growth and root development of cotton seedlings and specifically reduced plant height, total root length, total root surface area and volume, taproot length and taproot surface area. Exogenous melatonin seed soaking under normal and salt stress conditions can promote cotton seedling growth and root development, plant height, total root length, root volume and surface area and taproot length. However, soaking seeds in melatonin was unable to restore the root growth of cotton seedlings to normal levels under salt stress.

### 2.2. Effects of Exogenous Melatonin on the Root Antioxidant System of Cotton Seedlings under Salt Stress

Antioxidant enzyme activity levels are important indices of the ability of plants to remove peroxide. As shown in Figure 3, at 0 d, the SOD, POD and CAT activities under MT and MS treatments were higher than those under CK and S treatments, but there was no significant difference between CK and S or MT and MS treatments. At 5 d, the activities of SOD, POD and CAT under MT treatment were significantly higher than those under CK treatment. The SOD, POD and CAT activities under S treatment were significantly lower than those under CK treatment. The SOD, POD and CAT activities under MS treatment were higher than those under S treatment (*p* < 0.05). At 10 d, the POD and CAT activities under MT treatment were significantly higher than those under CK treatment. The SOD, POD and CAT activities under MS treatment were higher than those under S treatment (*p* < 0.05). At 15 d, SOD and POD activities under MT and CK treatments did not significantly differ. SOD, POD and CAT activities under S treatment significantly decreased by 29.68%, 20.89% and 46.47%, respectively, compared with CK treatment, while under MS treatment, they significantly increased by 14.45%, 20.75% and 59.61% compared with S treatment (*p* < 0.05). 

Non-enzymatic antioxidant substances (e.g., AsA and GSH) can scavenge free radicals and resist the negative effects of free radicals, so they can be used as an indicator of the ability of plants to respond to radicals. As shown in Figure 3D,E, at 0 d, the AsA and GSH contents under MT and MS treatments were significantly higher than those under CK and S treatments. At 5 d, there was no significant difference in AsA content between MT and CK treatments (*p* < 0.05). The AsA and GSH contents under MS treatment were significantly higher than those under S treatment. At 10 d, there was no significant difference in AsA content between MT and CK treatments (*p* < 0.05). The AsA content under S treatment was significantly higher than that under CK treatment (*p* < 0.05). The GSH content under MS treatment was higher than that under S treatment. At 15 d, there was no significant difference in AsA content between MT and CK treatments. The AsA content under S treatment was significantly higher than that under CK treatment, the GSH content under S treatment was significantly lower than that under MS treatment and the GSH content under MS treatment was significantly higher than that under S treatment (*p* < 0.05).

Similarly, H_2_O_2_ and MDA are important indices of membrane lipid peroxidation in plants. H_2_O_2_ content also increased with treatment time (Figure 3F). At 0 d of treatment, the content of H_2_O_2_ under MT and MS treatments was lower than under CK and S treatments, but there were no significant differences between these four treatments. At 5 d, H_2_O_2_ content under S treatment was significantly increased by 89.19% compared with CK treatment and by 54.70% compared with MS treatment (*p* < 0.05). At 10 d, S treatment significantly increased H_2_O_2_ content by 138% compared with CK treatment and by 46.96% compared with MS treatment (*p* < 0.05). At 15 d, S treatment significantly increased H_2_O_2_ content by 148% compared with CK treatment and by 59.03% compared with MS treatment (*p* < 0.05). As shown in Figure 3G, MDA content increased with treatment time. At 0 d of treatment, H_2_O_2_ content under MT treatment was significantly lower than that under CK treatment, and H_2_O_2_ content under MS treatment was significantly lower than under S treatment. At 5 d, S treatment significantly increased H_2_O_2_ content by 77.30% compared with CK treatment, and MS treatment significantly decreased H_2_O_2_ content by 14.68% compared with S treatment (*p* < 0.05). At 10 d, S treatment significantly increased H_2_O_2_ content by 67.54% compared with CK treatment, and MS treatment significantly decreased H_2_O_2_ content by 29.18% compared with S treatment (*p* < 0.05). At 15 d, MT treatment significantly decreased H_2_O_2_ content by 51.52% compared with CK treatment, and S treatment significantly increased H_2_O_2_ content by 65.96% compared with CK treatment and by 48.49% compared with MS treatment (*p* < 0.05). 

The above results indicated that soaking seeds in melatonin could reduce the accumulation of H_2_O_2_ and MDA in cotton roots under both normal conditions and salt stress. Thus, it promoted the activities of SOD, CAT and POD and increased the contents of AsA and GSH under salt stress, but the effect of melatonin on the root system under non-saline conditions (CK vs. MT) was not obvious. 

### 2.3. Effects of Exogenous Melatonin on Osmotic Substance Contents of Cotton Seedling Roots under Salt Stress

Osmotic regulation substances can alleviate osmotic damage to plant cells caused by salt stress. The trends in inorganic osmotic substances Na^+^ (Figure 4A) and K^+^ (Figure 4B) differed. At 0 d, the content of Na^+^ under MT and MS treatments was significantly higher than that under CK and S treatments; there was no significant difference between MT and MS treatments, and no significant difference between CK and S treatments. As treatment time increased to 5 d, the Na^+^ content under MT treatment was significantly lower than under CK treatment, the Na^+^ content under S treatment was significantly higher than under CK treatment and the Na^+^ content under MS treatment was significantly lower than under S treatment (*p* < 0.05). At 0 d, the content of K^+^ under MT and MS treatments was significantly higher than that under CK and S treatments. From 5 to 15 d, the K^+^ content under S and MS treatments gradually decreased with time. The K^+^ content under MT treatment was significantly higher than under CK treatment. Additionally, the K^+^ content under S treatment was significantly lower than under CK treatment, and the K^+^ content under MS treatment was significantly higher than that under S treatment (*p* < 0.05). 

The trends in the contents of soluble sugar (Figure 4C) and soluble protein (Figure 4D) were quite similar. At 0 d, the soluble sugar and protein contents under CK and S treatments were significantly higher than those under MT and MS treatments, but there was no significant difference between MT and MS or CK and S treatments. At 5, 10 and 15 d, MT treatment induced the highest contents of soluble sugar and protein, which were significantly higher than those under CK treatment. The contents of soluble sugar and protein under S treatment were the lowest and were significantly lower than those under CK treatment. The soluble sugar and protein contents under MS treatment were significantly higher than under S treatment (*p* < 0.05). Thus, soaking seeds in melatonin significantly promoted the soluble sugar, soluble protein and K^+^ contents in cotton seedling roots under both non-saline and salt stress conditions and reduced the accumulation of Na^+^ content.

### 2.4. Effects of Exogenous Melatonin on Root Hormones of Cotton Seedlings under Salt Stress

Plant hormones can regulate plant physiological processes and play an important role in plant growth. As shown in Figure 5, as the treatment time increased, the contents of IAA, ABA, GA and BR all changed to varying degrees, and the contents of IAA, GA and BR under different treatments increased gradually with time. At 0 d, the contents of IAA, GA and BR under MT treatment were significantly higher than those under CK treatment, and their contents under MS treatment were significantly higher than those under S treatment; however, there were no significant differences in IAA, GA or BR contents between MT and MS or CK and S treatments. At 5 d, the contents of IAA, GA and BR under MT treatment were higher than those under CK treatment, but not significantly. The contents of IAA, GA and BR under S treatment were significantly higher than those under CK treatment. The contents of IAA, GA and BR were the highest under MS treatment, and the contents of IAA and GA were significantly higher than those under S treatment (*p* < 0.05). At 10 d, the contents of IAA, GA and BR under S treatment were significantly higher than those under CK treatment. The contents of IAA, GA and BR under MS treatment were the highest and were significantly higher than those under S treatment (*p* < 0.05). At 15 d, the contents of IAA and GA under MT treatment were higher than those under CK treatment, but not significantly. The BR content under MT treatment was significantly higher than that under CK treatment, and the contents of IAA, GA and BR under S treatment were significantly higher than those under CK treatment. The contents of IAA and BR under MS treatment were significantly higher than those under S treatment (*p* < 0.05). Thus, salt stress significantly promoted IAA, GA and BR contents in cotton seedling roots, and exogenous melatonin increased the accumulation of IAA, GA and BR in the roots of cotton seedlings under non-saline and salt stress conditions.

At 0 d, the ABA content under MT and MS treatments was significantly lower than that under CK and S treatments, while there was no significant difference between CK and S or MT and MS treatments. At 5 d, the ABA content under MT treatment was significantly lower than that under CK treatment, the ABA content under S treatment was significantly higher than that under CK treatment, and the ABA content under MS treatment was significantly lower than that under S treatment (*p* < 0.05). At 10 d, the ABA content under S treatment was significantly higher than that under CK treatment. The ABA content under MS treatment was significantly lower than that under S treatment (*p* < 0.05), and the ABA content under MT treatment was significantly lower than that under CK treatment. At 15 d, the ABA content under MT treatment was lower than that under CK treatment, but not significantly. The ABA content under S treatment was significantly higher than that under CK treatment. The ABA content under MS treatment was significantly lower than that under S treatment (*p* < 0.05). Thus, exogenous melatonin significantly inhibited the accumulation of ABA content in roots. However, the inhibitory effect of exogenous melatonin was gradually weakened over time under non-saline conditions, and exogenous melatonin mainly had a significant effect on seedlings under salt stress. 

### 2.5. Effects of Exogenous Melatonin on Endogenous Melatonin Content and Root Activity of Cotton Seedlings under Salt Stress

Exogenous melatonin can affect the endogenous melatonin content of plants. As shown in Figure 6A, at 0 d of treatment, the endogenous melatonin content under MT and MS treatments was significantly higher than that under CK and S treatments, while there was no significant difference between MT and MS or CK and S treatments. As treatment time increased, the content of melatonin under different treatments increased slightly. At 5 d, the melatonin content under MT treatment was significantly higher than that under CK treatment, and the content of melatonin under MS treatment was significantly higher than that under S treatment (*p* < 0.05). At 10 d, MT treatment significantly increased the melatonin content by 13.16% compared with CK treatment, S treatment significantly decreased it by 9.73% compared with CK treatment, and MS treatment significantly increased the melatonin content by 9.69% compared with S treatment (*p* < 0.05). At 15 d, MT treatment significantly increased the endogenous melatonin content by 12.33% compared with CK treatment, and S treatment significantly decreased it by 9.35% compared with CK treatment; however, there was no significant difference in melatonin content between S and MS treatments. Thus, exogenous melatonin promoted endogenous melatonin content, while salt stress inhibited endogenous melatonin content. Additionally, the content of melatonin decreased gradually with salt stress duration. 

Root activity level has been shown to directly affect the growth of individual cotton seedlings and is therefore an important indicator of root development and vitality. As shown in Figure 6B, at 0 d, there was no significant difference in root activity between CK and S treatments, while root activity levels under MT and MS treatments were significantly higher than those under CK and S treatments, respectively. At 5, 10 and 15 d of treatment, the root activity under CK and MT treatments increased gradually with time, but the root activity under S and MS treatments first increased and then decreased over time. At 5 d, the root activity under S treatment significantly increased by 55.93% compared with CK treatment, but there was no significant difference between CK and MT treatments; additionally, the root activity under MS treatment significantly increased by 25% compared with S treatment (*p* < 0.05). At 10 d, the root activity under MT and S treatments significantly increased by 27.87% and 34.43%, respectively, compared with CK treatment, and the root activity under MS treatment was significantly increased by 24.39% compared with that of S treatment (*p* < 0.05). At 15 d, the root activity under MT and S treatments significantly increased by 14.29% and 18.57%, respectively, compared with CK treatment, and the root activity under MS treatment significantly increased by 12.28% compared with S treatment (*p* < 0.05). Thus, soaking cotton seeds in melatonin promoted root activity under non-saline and salt stress conditions.

### 2.6. Effects of Exogenous Melatonin on Taproot Tip Ultrastructure of Cotton Seedlings under Salt Stress

The root tip is the most active site of root growth and the absorption of water and nutrients in crops. To further study the effects of exogenous melatonin on the root tip structure of cotton seedlings under salt stress, we observed the ultrastructure of the cotton root tips.

#### 2.6.1. Differences in Root Epidermal Cell Characteristics of Cotton Seedlings among Treatments

The root epidermis is the main water-absorbing structure of crops. As shown in Appendix A, the epidermal cell structures differed among treatments. Under CK treatment (Appendix A), the epidermal cells were intact, without cytoplasmic wall separation, and the structures of the mitochondria, Golgi apparatus, endoplasmic reticulum and nucleus were intact. Under MT treatment (Appendix A), epidermal cells were also intact, without cytoplasmic wall separation and with a complete nuclear structure, clear nuclear membrane and clearly visible mitochondrial edge contours and internal ridges. However, Golgi apparatus and endoplasmic reticulum organelles were more abundant and more numerous under MT treatment than CK treatment. Under S treatment (Appendix A), the epidermis was damaged, and the severe separation of the plasma wall, nuclear nucleolus cleavage and nuclear membrane rupture occurred in epidermal cells. Additionally, the inner ridge of the mitochondria was not obvious, and mitochondrial membrane cleavage occurred. Compared with CK treatment, S treatment reduced the number of mitochondria and basically ruptured both the endoplasmic reticulum and the Golgi apparatus. Under MS treatment (Appendix A), the epidermis was slightly damaged, the epidermis cells were slightly separated from the plasma wall and the nuclear structure was complete. Compared with S treatment, under MS treatment, the mitochondrial internal ridge was not obvious, but the structure was complete, and the number was increased compared with that under S treatment. Additionally, the structure of the endoplasmic reticulum and Golgi apparatus was relatively complete.

#### 2.6.2. Differences in Root Cortex Cell Characteristics of Cotton Seedlings among Treatments

The cortex is the main structure for lateral transport and the storage of nutrients in crop roots. As shown in Figure 6, the structure of cortical parenchyma cells also differed among treatments. Under CK treatment (Figure 7A–F), cortical cells were arranged completely, and no plasma wall separation occurred in individual cortical cells. The structures of the mitochondria, Golgi apparatus, endoplasmic reticulum and nucleus were intact. Under MT treatment (Figure 7G–L), cortical cells were completely and closely arranged, with unseparated cell walls and complete nuclear and mitochondrial structures, and the number of Golgi apparatus and endoplasmic reticulum organelles increased compared with that under CK treatment. Under S treatment (Figure 7M–R), cortical cells were loosely arranged, with serious separation of the cytoplasm from the cell wall, severe invagination of the cell membrane, nuclear lysis, nuclear membrane rupture, few mitochondria occurring in cells and the occurrence of cracked mitochondria. Compared with CK treatment, the number of mitochondria under S treatment decreased, and endoplasmic reticulum and Golgi apparatus organelles were basically cracked and had somewhat disappeared. Under MS treatment (Figure 7S–X), epithelial cells were arranged loosely compared with CK treatment, but compared with S treatment, cells were arranged more closely, cortical cells were more complete, with slight plasmolysis of epidermis cells, and the nucleus structure was intact. Additionally, the mitochondrial structure was complete, and the number of mitochondria was greater than that under S treatment, with more endoplasmic reticulum and Golgi apparatus organelles, higher structural integrity and a greater abundance of organelles.

#### 2.6.3. Differences in Root Phloem Cell Characteristics of Cotton Seedlings among Treatments

The phloem is located inside the vascular bundle, and the phloem parenchyma cells mainly transport nutrients. As shown in Appendix A, the structure of phloem parenchyma cells differed to varying degrees among treatments. Under CK treatment (Appendix A), these cells were intact; no plasma wall separation occurred in individual cells; and the structures of the mitochondria, Golgi apparatus, endoplasmic reticulum and nucleus were intact. Under MT treatment (Appendix A), the cells were intact and arranged more closely compared with CK treatment, and there was no separation of the plasma wall in cells. The structures of the nucleus and mitochondria were intact, but the mitochondria, Golgi apparatus and endoplasmic reticulum organelles were more numerous than under CK treatment. Under S treatment (Appendix A), cells were intact, without cytoplasmic wall separation and with a complete nuclear structure, and only the mitochondria were retained. The inner ridge of the mitochondria was not obvious, and the mitochondrial membrane was cleaved. Compared with CK treatment, the number of mitochondria under S treatment decreased, and the endoplasmic reticulum and Golgi apparatus were basically cleaved and had disappeared somewhat. Under MS treatment (Appendix A), the cells were intact, but compared with S treatment, the cells were more closely arranged, without plasma wall separation and with complete nuclear and mitochondrial structures, and cells were more numerous than under S treatment. The endoplasmic reticulum and Golgi apparatus structures were also complete, and the organelles were more abundant than under S treatment.

Thus, after salt stress treatment, cotton root cells were arranged loosely with fewer organelles and only mitochondria retained, and the internal ridge of the mitochondria was not obvious, with cleavage occurring and the endoplasmic reticulum and Golgi apparatus structures disappearing. Under non-saline conditions and salt stress conditions with melatonin, cotton seedling root cell organelles were more abundant, especially endoplasmic reticulum and Golgi apparatus organelles, which are critical components of protein and lipid synthesis and transport. Accordingly, exogenous melatonin can promote protein and lipid metabolism in cotton seedlings to increase resistance to salt-induced damage.

### 2.7. Transcriptomic Analysis of Cotton Seedling Root Systems under Melatonin and Salt Stress

To explore how exogenous melatonin alters the response of cotton roots to salt stress at the molecular level, at 48 h into treatment, three biological replicates from the CK, MT, S and MS treatments were sampled for transcriptome and metabolome analysis, for 12 samples in total.

#### 2.7.1. Functional Analysis of Gene Expression Changes in Roots of Melatonin-Treated Cotton Seedlings under Salt Stress

By comparing the RNA-seq data between the treatment groups, i.e., S vs. CK, MT vs. CK, MS vs. S and MS vs. MT (Figure 8), log2 (fold change (FC)) and *p* < 0.05 thresholds were applied as the standard for identifying differentially expressed genes, and statistical methods were used to analyze the differentially expressed genes under different treatments. As shown in Figure 8A, compared with CK treatment, a total of 2519 genes were differentially expressed under S treatment, including 1392 up-regulated genes and 1127 down-regulated genes. Compared with CK treatment, MT treatment induced a total of 214 differentially expressed genes, including 117 up-regulated genes and 97 down-regulated genes. Compared with S treatment, a total of 347 differentially expressed genes were observed under MS treatment, including 237 up-regulated genes and 110 down-regulated genes. Compared with MS treatment, there were 2701 differentially expressed genes under MT treatment, including 1602 up-regulated genes and 1099 down-regulated genes. As shown in Figure 8B, 888 genes were differentially expressed in both the CK vs. S and S vs. MS comparisons, and 5997 and 976 genes were uniquely expressed under the S and MS treatments, respectively. There were 442 genes uniquely expressed in both the CK vs. MT and MT vs. MS comparisons, and 6399 and 534 genes were uniquely expressed in the MS and CK treatments, respectively. In the comparisons among treatments, the number of up-regulated genes was higher than that of down-regulated genes. 

To conduct an in-depth analysis of the functions of differentially expressed genes, we selected the most significant KEGG terms (*p* < 0.05) and conducted KEGG pathway analysis on the differentially expressed genes between treatments, as summarized in the KEGG enrichment cycle diagram in Figure 8C–F and Appendix A. Compared with CK treatment (Figure 8C, Appendix A), the differentially expressed genes under MT were mainly enriched in the phenylpropanoid biosynthesis (ko00940), photosynthesis (ko00195), biosynthesis of secondary metabolites (ko01110), glycerophospholipid metabolism (ko00564) and biosynthesis of amino acids (ko01230) pathways, in which a gene encoding a MYB transcription factor was significantly up-regulated. Compared with CK treatment (Figure 8D, Appendix A), the differentially expressed genes under S treatment were mainly enriched in the biosynthesis of secondary metabolites (ko01110), phenylpropanoid biosynthesis (ko00940), plant hormone signal transduction (ko04075), flavonoid biosynthesis (ko00941), MAPK signaling (ko04016), glyoxylate and dicarboxylate metabolism (ko00630), fatty acid degradation (ko00071) and carbon metabolism (ko01200) pathways, in which the same transcription factor encoding MYB was significantly down-regulated. In the MAPK signaling pathway, transcription factors encoding WRKY33 were significantly down-regulated, and in the plant hormone signal transduction pathway, transcription factors encoding the ethylene-related transcription factors ERF1 and TGA were significantly down-regulated. Compared with S treatment (Figure 8E, Appendix A), the differentially expressed genes under MS treatment were mainly enriched in the biosynthesis of secondary metabolites (ko01110); MAPK signaling (ko04016); fatty acid degradation (ko00071); galactose metabolism (ko00052); alanine, aspartate and glutamate metabolism (ko00250); phenylpropanoid biosynthesis (ko00940); glyoxylate and dicarboxylate metabolism (ko00630); pantothenate and CoA biosynthesis (ko00770) and plant hormone signal transduction (ko04075) pathways. Among these, genes encoding MYB transcription factors were significantly down-regulated, and WRKY transcription factors were enriched in the MAPK signaling pathway. Related genes were significantly up-regulated, including genes encoding ERF1 transcription factor, and genes encoding TGA and ERF1/2 transcription factors in the plant hormone signal transduction pathway were significantly up-regulated. Compared with MT treatment (Figure 8F, Appendix A), the differentially expressed genes under MS treatment were mainly enriched in the biosynthesis of secondary metabolites (ko01110), phenylpropanoid biosynthesis (ko00940), plant hormone signal transduction (ko04075), fatty acid degradation (ko00071), arginine and proline metabolism (ko00330), glycerophospholipid metabolism (ko00564), flavonoid biosynthesis (ko00941) and tryptophan metabolism (ko00380) pathways. Among these, the genes encoding MYB transcription factors were significantly down-regulated. In the MAPK signaling pathway, genes encoding WRKY33 transcription factors were significantly down-regulated, while genes encoding ERF and TGA transcription factors in the plant hormone signal transduction pathway were both up-regulated and down-regulated. The KEGG enrichment analysis showed that compared with CK treatment, the differentially expressed genes under S treatment were mainly enriched in the phenylpropanoid biosynthesis, plant hormone signal transduction, fatty acid degradation and carbon metabolism pathways, and a large number of genes encoding MYB, WRKY, ERF and TGA transcription factors were significantly down-regulated. Thus, salt stress likely inhibited the root development of cotton seedlings by affecting the MAPK signaling pathway, hormone signal transduction and transcription factor-related gene expression. The differentially expressed genes under MS treatment were mainly enriched in the MAPK signaling, plant hormone signal transduction, amino acid synthesis and metabolism pathways, and a large number of genes encoding MYB, WRKY, ERF and TGA transcription factors were significantly up-regulated in the CK vs. MT and S vs. MS comparisons. These results indicate that exogenous melatonin can mitigate the inhibitory effect of salt stress on plant root development by affecting plant hormone signal transduction and up-regulating the expression of MYB, WRKY, ERF and TGA transcription factors. Notably, compared with MS treatment, the expression levels of ERF and TGA transcription factors under MT treatment were not down-regulated in the CK vs. S comparison nor up-regulated in the CK vs. MT or S vs. MS comparisons, but they were up-regulated and down-regulated in the MS vs. MT comparison, indicating that salt stress and melatonin could affected each other. Thus, salt stress can affect the regulation of melatonin in plants.

#### 2.7.2. Quantitative Analysis of the Effect of Exogenous Melatonin on Differentially Expressed Genes in Cotton Seedling Roots under Salt Stress

To verify the accuracy of the differentially expressed genes identified by RNA-seq, eight differentially expressed genes (Gh_A12G288400, Gh_A13G037600, Gh_Contig00024G000100, Gh_D05G131400, Gh_D06G191100, Gh_D08G239500, Gh_D11G139300 and Gh_D13G014400) were randomly selected from the transcriptome database for confirmation by qRT-PCR (Appendix A). The trends identified in the transcriptome data were basically consistent with those observed in the qRT-PCR analysis data, indicating that the transcriptome data showed high reliability. These results also suggest that exogenous melatonin can regulate the root growth of cotton seedlings by regulating the expression of auxin, cytokinins, abscisic acid and endogenous melatonin-related genes.

### 2.8. Metabolomic Effects of Exogenous Melatonin on the Root Systems of Cotton Seedlings under Salt Stress

#### 2.8.1. Analysis of Differentially Accumulated Metabolites in Roots of Cotton Seedlings under Salt Stress and Exogenous Melatonin Treatment

The analysis of the signaling pathways of differentially accumulated metabolites can reveal the main biochemical and signal transduction pathways involved in the differential accumulation of those metabolites. As shown in Figure 9A, compared with CK treatment, the metabolites accumulated under MT treatment were significantly enriched for flavonoids, sugars and alcohols, mainly including glucarate O-phosphoric acid, d-arabinono-1,4-lactone and rutinose; indoles; alkaloids; melatonin; organic acids (γ-aminobutyric acid, isocitric acid); lipids (e.g., lysophosphatidylcholines, free fatty acids); and terpenoids such as gossypol. Differentially accumulated metabolites were mainly enriched in association with terpenoid material synthesis, the phosphatidyl inositol signaling system, alanine, tyrosine metabolism, biotin, inositol phosphate metabolism, arginine and proline metabolism, flavonoid synthesis, the TCA cycle, pantothenic acid and CoA biosynthesis, arginine synthesis, alanine, glutamic acid and aspartic acid metabolism, ascorbic acid salt metabolism and fatty acid synthesis metabolic pathways. Compared with CK treatment (Figure 9B), the metabolites under S treatment were significantly enriched in amino acids and their derivatives l-cysteine, serine, l-glutamine, l-proline, l-arginine, S-(methyl)glutathione and other amino acids. The enriched metabolites included flavonoids, sugars and alcohols (d-fructose-6-phosphate, 3-phospho-d-glycolic acid, d-glucurono-6,3-lactone, d-xylonic acid, d-glucose 6-phosphate and gluconic acid, among others). The differentially accumulated metabolites were mainly enriched in association with glucuronic acid conversion, secondary metabolite synthesis, amino acid synthesis, ABC transporter, antioxidant synthesis, glycerol phospholipid metabolism, arginine synthesis, arginine and proline metabolism, carbon metabolism, phosphoinositol metabolism and isoflavone synthesis pathways, among other metabolic pathways. Compared with S treatment (Figure 9C), the metabolites accumulated under MS treatment were significantly enriched for flavonoids, saccharides raffinose, free fatty acids, lysophosphatidylethanolamine, lysophosphatidylcholines amino acids and their derivatives. The differentially accumulated metabolites were mainly enriched in association with the metabolic pathways of carbon metabolism, antioxidant synthesis, amino and nucleotide sugars, starch and sucrose, glycerophospholipid and sphingolipid. Compared with MT treatment (Figure 9D), the metabolites under MS treatment were significantly enriched in sugars and alcohols (glucose-1-phosphate, N-acetyl-d-mannose, gluconic acid, sorbitol-6-phosphate, among others); flavonoids; amino acids and their derivatives (serine); alkaloids (ethanolamine phosphate); phosphatidylcholine and other substances. The differentially accumulated metabolites were mainly enriched in association with biotin metabolism, ABC transporter, galactose metabolism, flavonoid synthesis, antioxidant synthesis, secondary metabolite synthesis and amino acid synthesis pathways. These results showed that salt treatment accelerated lipid metabolism and carbon metabolism and promoted amino acid synthesis in the root system of cotton seedlings. Exogenous melatonin mainly promoted the synthesis and metabolism of antioxidants, the synthesis of ABC transporters, the synthesis of melatonin and hormones, the synthesis of amino acids, and the metabolism of membrane lipids and sugars for energy conversion, thus improving the tolerance of cotton seedling roots to salt stress.

#### 2.8.2. Combined Transcriptomic and Metabolomic Analysis

Based on the KEGG analysis, we analyzed the DEGs related to ROS clearance, plant hormone signal transduction, melatonin synthesis and substance transport (Figure 10A), revealing that six DEGs were related to ROS clearance. Three of these were associated with the POD pathway (Gh_A12G007900, Gh_A05G154800, Gh_A12G176900) and two with the CAT pathway (Gh_A02G208700, Gh_D01G104300), while the remaining one was associated with the GSH pathway (Gh_A04G117100). Twenty DEGs were associated with plant hormone signal transduction and specifically associated with five subpathways, including eight DEGs associated with the auxin subpathway, two with the CTK (cytokinin) subpathway, four with the GA subpathway, four with the ABA subpathway and two with the brassinosteroid (BR) pathway. The analysis of the S vs. MS comparison showed that the auxin protein gene Aux/IAA, auxin small RNA gene SAUR and auxin response gene GH3 responded to auxin during auxin signal transduction, and four auxin protein genes (Aux/IAAs) were significantly up-regulated. Two genes encoding SAUR and two encoding GH3 were down-regulated. For the CTK pathway, both up- and down-regulated DEGs genes included cytokinin receptor (AHK) genes. For GA signal genes, MT treatment up-regulated two genes encoding GA receptor (GID1). For ABA signaling, four genes were up-regulated, mainly encoding ABA response element binding factor (ABF). For BR signaling, MT treatment induced the up-regulation of two genes encoding Br-associated receptor kinase (BZR). In addition to hormone signal transduction, six genes related to melatonin and tryptophan synthesis were up-regulated (Gh_A08G128700, Gh_A12G266800, Gh_D08G239500, Gh_D09G122800, Gh_A02G091100 and Gh_D06G170900). Four genes encoding ABC transporters (Gh_A07G239500, Gh_A08G172900, Gh_D11G383100 and Gh_A02G052100) were up-regulated.

As shown in Figure 10B and Appendix A, each gene had multiple significantly associated metabolites. The first metabolites that were significantly associated with POD genes (Gh_A12G007900, Gh_A05G154800 and Gh_A12G176900) were ferulic acid (MWS0014), isoferulic acid (PME0422), dihydroxyacetone phosphate (Zmzn000078), sedoheptulose (Hmcn000192), nicotine (MWS1478), p−coumaroylquinic acid−4′−O−glucuronide (PMB3055) and luteolin−4′−O−glucoside (Hmpp003270). The metabolites significantly associated with CAT genes (Gh_A02G208700 and Gh_D01G104300) were 5−glucosyloxy−2−hydroxybenzoic acid methyl ester (Lmmn003663), nicotinamide (MWS0133), N-monomethyl−l−arginine (Zmjp000182), l−arginine (MWS) McE119), l−aspartic acid (MWS0219), 5,7,3′−trihydroxy−4′−methoxyflavone (MWSHY0009), O−acetylserine (MWS1050), indole 3−acetic acid (PME1651), lysophosphatidylethanolamine (Lmhp008440), glucose−1−phosphate (MWS1090), l−serine (pme0010), gluconic acid (pme0534) and 2−(formylamino)benzoic acid (pme3083), among others. The metabolites associated with GSH genes (Gh_A04G117100) were p−coumaroylquinic acid−4′−O−glucuronide (pmb3055) and luteolin−4′−O−glucoside (Hmpp003270). As shown in the correlation heat map in Appendix A, the differentially expressed genes and differentially accumulated metabolites detected by transcriptome and metabolome analyses were mainly enriched in association with amino acids, lipids, flavonoids, acids and sugars. For example, detected amino acids were mainly enriched for arginine (MWSmce119, Zmjp000182, Zmdp000292, PME3388, MWS5164 and NK10251888). The genes associated with arginine were MSTRG.22382, Gh_D04G076400, Gh_A11G001000 and Gh_A03G239500, among others. The enriched lipids included glyceride (PMB0296), lysophosphatidylcholine (PMB0863, Lmhp007598, PMB0854, PMP001270 and PMP001251), lysophosphatidylethanolamine (Lmhp009034, PMB0874), glyceride (Lmhp009773), Gh_D13G028000, Gh_A02G174600, Gh_D03G031800, Gh_D02G243900 and Gh_D11G135500. The flavonoid metabolites included MWSHY0009, MWSHY0069, PMP000592 and MWS0918, and the genes significantly associated with them were Gh_D05G182800, Gh_D05G154700 and Gh_D08G195200. The identified acid substances mainly included abscisic acid (Lmtn004049), and its related genes included Gh_D12G275400, Gh_A12G288400, Gh_A10G050100, Gh_A03G207200 and Gh_A05G178100. The identified saccharides included glucuronic acid (PME0534), d−galacturonic acid (MWS1189), d−fructose−6−phosphate (MWS2442), d−fructose (MWS1164), d−glucose−6−phosphate (MWS0866) and d−glucuronic acid (PME3705). The genes significantly associated with these metabolites were Gh_D02G047500, Gh_D13G262800, Gh_D04G178200, Gh_D10G191000, Gh_D11G151600, Gh_D09G216700 and MSTRG.1567, among others. Overall, exogenous melatonin was able to alter the expression of antioxidant enzyme (POD, CAT, GSH) genes, hormone signaling pathway genes, endogenous melatonin-related genes and material transport-related genes to induce the synthesis of arginine, glucose and fructose, substance, phospholipids, abscisic acid and flavonoids in cotton roots. Thus, exogenous melatonin promoted root development under salt stress.

## 3. Discussion

### 3.1. Effect of Exogenous Melatonin on the Antioxidant System of Cotton Seedling Roots under Salt Stress

When plants are injured by salt stress, they respond through a series of salt tolerance mechanisms that can remove harmful ROS and free radicals through increased antioxidant enzyme activity [33]. The antioxidant enzyme system in plants mainly includes SOD, POD and CAT, which can regulate the oxidative stress damage that occurs in plants [34]. In previous research on the effects of salt stress on the physiological characteristics of soybean [35] and wheat [36], it was found that the activities of SOD, POD and CAT increased under low salt concentrations. As salt stress treatment duration and concentration increased, SOD activities increased first and then decreased, while POD and CAT still showed upward trends. At the same time, plant antioxidant enzyme activity did not reflect this pattern of a continued increase. Studies have shown that exogenous melatonin can promote the activity of antioxidant enzymes in tomato, oilseed rape, Arabidopsis thaliana and other plants under stress, thus improving the tolerance of plants to stress [37,38].The results of the present study showed that salt stress treatment could indeed improve the POD and CAT activities of cotton seedling roots within a short time period; however, as salt stress duration increased, POD activity and CAT activity decreased significantly and increased, respectively. Thus, we hypothesized that ROS signal expression may differ among species, explaining the decline in POD activity in some species. After soaking the seeds in melatonin, the exogenous application of melatonin promoted the POD and CAT activities of cotton seedling roots under both non-saline conditions and salt stress, but the effect of soaking the seeds in melatonin on the POD activities of cotton seedling roots under non-saline conditions was not significant. This may be due to the fact that the accumulation of ROS under normal conditions is insufficient to significantly up-regulate the expression of antioxidant-related genes.

GSH is a ubiquitous oligopeptide in plant cells and an important antioxidant in the ASA–GSH cycle pathway, which plays a critical role in scavenging excess ROS in plants, and the two usually act synergistically [39,40]. In cucumber and maize seedling leaves, melatonin treatment increased both the concentration of GSH and the tolerance of plants to salt stress [41,42]. In the present study, as the salt stress duration increased, the content of GSH in the root system of the cotton seedlings first increased and then decreased. Cotton is a salt-tolerant crop, and the initial increase in GSH was conducive to the scavenging of free radicals. However, as the stress duration continued, the GSH content began to decline when large amounts of reactive oxygen species were produced. Soaking seeds in melatonin significantly promoted GSH content under salt stress but had no significant effects on the root system of cotton seedlings under non-saline conditions, and even reduced the content of GSH, which was hypothesized to be because GSH participates in root cell division and thus caused its content to decline [43].

MDA is the metabolic end product of lipid peroxidation. MDA content reflects the level of free radicals in plant tissues and the degree of lipid peroxidation caused by free radicals [44]. Salt stress can induce a series of secondary reactions to plant osmotic stress and ion stress, which greatly increases the content of MDA and ROS in plants [45,46]. Melatonin is recognized as the strongest free radical scavenger and can scavenge both free radicals and ROS in plants [47]. In the present study, the MDA content of cotton seedling roots increased with the salt stress duration, which was consistent with previous studies [48]. Soaking seeds in melatonin was able to reduce the MDA content in cotton seedling roots under both non-saline and salt stress conditions, but only significantly under salt stress. We hypothesized that because seedlings produced fewer free radicals under normal conditions, there were fewer signal molecules regulating the function of melatonin, and the ability of melatonin to scour free radicals was relatively weak; thus, the effect of melatonin was not obvious under non-saline conditions.

### 3.2. Effect of Exogenous Melatonin on Osmotic Substances in Cotton Seedling Roots under Salt Stress

When plants are subjected to salt stress, a series of complex osmotic stress and osmotic responses occur. For example, salt stress can lead to an abundance of Na^+^ entering plant cells, and high concentrations of Na^+^ damage cell solute enzymes, resulting in ion stress [49]. To maintain the normal physiological metabolism of intracellular water under salt stress, cells reduce intracellular water potential by regulating organic osmotic substances and inorganic ions and promote the transmembrane transport of water in a direction conducive to cell growth. Under salt stress, plant soluble sugar and protein first increased and then decreased [50]. Melatonin treatment can increase the contents of soluble sugar and protein, participate in osmotic regulation and alleviate the inhibitory effect of stress on plant growth [51]. The present study showed that as salt stress duration increased, cotton seedling root soluble sugar and soluble protein contents increased first and then decreased, Na^+^ content increased and K^+^ content continuously decreased. This pattern may be explained by the accumulation of Na^+^ under salt stress, which inhibits the absorption of K^+^ by roots and then leads to cell membrane damage, resulting in the instability of the synthetic metabolic pathways of sugars and proteins. Exogenous melatonin significantly promoted the production of soluble sugar, soluble protein and K^+^ content in the roots of cotton seedlings under non-saline and salt stress conditions and inhibited the accumulation of Na^+^ content, which was consistent with previous studies [52]. This also indicates that exogenous melatonin can reduce the accumulation of Na^+^ content, maintain the integrity of plant cell membranes and promote the production of both soluble sugar and soluble protein in plants.

### 3.3. Effect of Exogenous Melatonin on Root Hormones of Cotton Seedlings under Salt Stress

A critical function of the root system in plants is the transport of water and nutrients [46]. Melatonin occurs widely among plants and animals, and the discovery of a melatonin receptor in plants in 2018 suggested that melatonin can also be considered a plant hormone [53]. Salt stress is among the main environmental stresses limiting plant growth. Many studies have shown that plant hormones not only control the growth and development of plants under normal conditions, but also mediate the response of plants to various environmental stresses, including salt stress, so as to regulate plant growth [54]. Among the various plant hormones, ABA and ethylene are considered to function as stress response hormones. Other factors, such as IAA, GA, CTKs and BRs, are also classified as growth-promoting hormones [55]. Under salt stress, endogenous ABA levels have been observed to increase rapidly and activate the synthesis of related protein kinases [56]. IAA plays an important role in plant development, affecting root morphology, inhibiting root elongation and increasing the number of lateral roots [57]. The inhibition of root growth by salt may be an adaptive mechanism for plant survival, which is closely related to decreased IAA accumulation [58]. GA plays an important role in promoting cell elongation and growth [59]. Salt stress reduced GA levels and inhibited plant growth [60]. BR plays an important role in plant adaptation to salt stress, and exogenous BR can alleviate the inhibition of plant growth by salt stress [57,61,62]. Melatonin is an important regulator of the expression of plant hormone-related genes and plays an important role in auxin carrier proteins and the metabolism of auxin, gibberellin, cytokinin and abscisic acid. In the present study, salt stress reduced the accumulation of endogenous melatonin in the roots of cotton seedlings and promoted increases in IAA, GA, BR and ABA contents in cotton seedling roots, in contrast with previous studies. Thus, there appears to be a dynamic balance between the stress hormone ABA and the growth hormones IAA, GA and BR in regulating salt stress signals and the stress response and controlling root growth, and the synthesis and metabolism of hormones under salt stress may also depend on the plant growth environment, development stage and plant species. Melatonin seed treatments significantly increased endogenous melatonin, IAA, GA and BR contents and reduced the accumulation of ABA content in cotton seedling roots under both non-saline and salt stress conditions. However, after exogenous melatonin was applied under non-saline conditions, the reduction effect of ABA gradually weakened as the stress duration continued, which may be due to the increase in ABA content under salt stress. However, the change in ABA content was not obvious under non-saline conditions, as melatonin also reduced the synthesis of ABA-related proteins [56]. 

### 3.4. Effect of Exogenous Melatonin on Root Microstructure of Cotton Seedlings under Salt Stress

Plants accumulate high levels of ROS under salt stress, leading to changes in cell structure. The most typical symptom of these changes is cell membrane lysis, while more specific changes include increased cell vacuolation, the deformation of nuclear chromatin and membrane damage [63]. Under Cd stress, the ultrastructural changes of cotton root tip cells mainly occurred in the plasma membrane, nuclear membrane, vacuoles and nucleoli. As the Cd concentration increased and formed a harmful solute, the number of vacuoles in the root tip cells continually increased, and the number of nucleoli increased [64]. The present study showed that when cotton seedling roots were subjected to salt stress, cortex cells were the first to experience stress injury, and their damage was the most serious, resulting in severe plasma wall separation; cell membrane cleavage; nuclear deformation; nucleolar material increases; decreases in the number of mitochondria, Golgi apparatus and endoplasmic reticulum organelles; and mitochondria with a less obvious inner ridge. The effect of melatonin on the nuclear membrane and nucleoli was not obvious under normal conditions, but the number of endoplasmic reticulum and Golgi apparatus organelles increased significantly, which might be because melatonin promoted the synthesis and metabolism of proteins and lipids in cotton seedling roots. The addition of melatonin to plants under salt stress can slow down the separation of the plasma wall; maintain the integrity of cellular structure; and increase the number of mitochondria, endoplasmic reticulum and Golgi apparatus organelles compared with control conditions. Compared with cortical cells, the phloem cells were more intact under each treatment, and the cytoplasmic wall separation occurred to some degree under salt treatment. Additionally, only mitochondria and nuclei were observed in phloem cells. Under control conditions and under salt stress, the application of melatonin was able to promote increases in the abundance of root mitochondria, endoplasmic reticulum and Golgi apparatus organelles. This also suggested that the application of melatonin, by increasing the number of mitochondria, could promote the accumulation of endogenous melatonin and, by increasing the number of Golgi apparatus and endoplasmic reticulum organelles, could promote the synthesis of protein and lipid transport and thus metabolism so as to reduce the damage to plants caused by salt stress. 

### 3.5. Effects of Exogenous Melatonin on Root Metabolic Pathways of Cotton Seedlings under Salt Stress

Based on the above results, exogenous melatonin can eliminate ROS produced by salt stress by regulating the antioxidant system, regulate the content of plant hormones, promote protein and lipid synthesis and metabolism by increasing the number of endoplasmic reticulum and Golgi apparatus organelles in root cells and promote root development in cotton seedlings. To further explore the effect of melatonin on cotton roots, we conducted transcriptomic and metabolomic analyses on cotton roots under different treatments. The transcriptome results showed that exogenous melatonin could regulate the root growth of cotton seedlings by affecting plant hormone signal transduction, amino acid synthesis and metabolism and the expression of some transcription factors under salt stress. The exogenous application of melatonin promoted the overexpression of endogenous melatonin synthesis genes and thus increased the content of endogenous melatonin. Exogenous melatonin promoted plant signal transduction and MAPK signal transduction pathways, mainly through auxin, the overexpression of cytokinin synthesis genes and the down-regulation of abscisic acid receptor genes, suggesting that exogenous melatonin may promote plant growth under stress by regulating the expression of plant hormone-related genes. Similarly, Arnao and Hernandez-Ruiz found that melatonin can positively and negatively interact with other plant hormones (including IAA, BR, GA, ET, JA and ABA) [65]. The results of this study are consistent with previous results. By comparing non-saline and salt stress conditions with melatonin treatment conditions, we found that salt stress mediated the effect of melatonin on cotton root systems, and significant enrichment of the melatonin tryptophan metabolic pathway was observed under salt stress, which may have been caused by the increase in endogenous melatonin synthesis induced by salt stress [66]. The metabolomic analysis results showed that after melatonin application, metabolites were mainly enriched for lipid metabolism, carbohydrate metabolism, and amino acid metabolism and synthesis pathways. These results suggest that exogenous melatonin promotes an increased abundance of mitochondria and endoplasmic reticulum and Golgi apparatus organelles in cells and accelerates the amino acid, lipid and carbohydrate metabolism process, thereby promoting cotton root growth and development.

### 3.6. Regulation Mode of Exogenous Melatonin on Root Growth of Cotton under Salt Stress

Based on the results of the present study, we developed a simplified model to describe the effect of exogenous melatonin on roots under salt stress. As shown in Figure 11, exogenous melatonin can promote increases in SOD, CAT and POD activities and AsA and GSH contents in the roots of cotton seedlings under salt stress, which leads to a decrease in the accumulation of H_2_O_2_ and MDA. Melatonin can also promote the production of soluble sugar, soluble protein and K^+^ and reduce the accumulation of Na^+^. Exogenous melatonin also promoted both endogenous melatonin content and root activity as well as the synthesis of IAA, GA and BR. The melatonin treatment also reduced the accumulation of ABA and induced the expression of various genes and transcription factors, such as WRKY, ERF, MYB and TGA. The application of melatonin can improve the expression of endogenous melatonin-related genes and thus the level of melatonin, resulting in a close arrangement of cotton root cells and an increase in the number of organelles such as mitochondria and Golgi apparatus and endoplasmic reticulum organelles. Exogenous melatonin can significantly enrich metabolites associated with carbon metabolism; antioxidant synthesis; amino acid, sugar and nucleotide sugar metabolism; starch and sucrose metabolism; glycerophospholipid metabolism and sphingolipid metabolism pathways. Thus, exogenous melatonin can alleviate salt stress in cotton roots by improving the activity of antioxidant enzymes, promoting the synthesis of osmotic regulatory substances, activating transcription factors and coordinating complex hormone signal transduction pathways and hormone synthesis. In general, exogenous melatonin can improve the salt tolerance of cotton seedlings by regulating their root physiology, cell structure and gene expression under salt stress.

## 4. Materials and Methods

### 4.1. Plant Materials

The cotton (*Gossypium hirsutum* L.) variety ‘Guoxin No. 9,’ which was provided by the Guoxin Rural Technical Service Association, Hebei Province, China, was used as plant material for this experiment. Melatonin was obtained from Sigma Co., Ltd. (St. Louis, MO, USA). Melatonin was diluted with 95% ethanol to 200 mmol∙L^−1^ as a stock solution and stored at −20 °C for later use. The nutrient solution for the test was standard Hoagland nutrient solution.

### 4.2. Methods

#### 4.2.1. Experimental Treatment

The experiment was conducted in the greenhouse facilities of Hebei Agricultural University, Baoding City, Hebei Province, China. Cotton seeds of uniform size were selected and then disinfected with 75% ethanol. After wiping away their surface moisture, the seeds were soaked in 0 μmol∙L^−1^ (distilled water) or 10 μmol∙L^−1^ [32] melatonin solution for 24 h at 25 °C in the dark. The soaked seeds were placed flat on a tray with wet towels. When the radicle of each seed reached 1 cm, it was fixed onto a foam board. The foam board was previously sterilized and punctured with holes with a diameter of 0.3 cm. The fixed seeds were placed in a light incubator for hydroponic cultivation (temperature controlled at 25 °C, photoperiod at 14 h/10 h, relative humidity constant at 60%, light intensity at 600 μmol∙m^−2^∙s^−1^). When the cotton seedlings had two flat cotyledons, they were moved to an incubator with external dimensions of 485 × 355 × 245 mm. At the two-true-leaf stage, seedlings of both non-treated and melatonin-treated seeds were subjected to the following treatments: 0 μmol∙L^−1^ NaCl and 150 mmol∙L^−1^ NaCl [67] (Table 1).

Each treatment was applied to three replicates, and the nutrient solution was replaced every 3 days. Oxygen was supplied at 8:00 and 10:00 for 1 h. The temperature was maintained at 28 °C/25 °C, with 7% relative humidity, a 14 h/10 h photoperiod and a light intensity of 600 μmol∙m^−2^∙s^−1^.

#### 4.2.2. Determination Method

##### Determination of Antioxidant Enzyme Activity, Contents of Osmotic Regulatory Substances and Root Activity

At 0, 5, 10 and 15 d after treatment, 0.2 g samples were collected from three replicates under each treatment, and root physiology indices of the cotton seedlings were recorded, including antioxidant enzyme activity, contents of osmotic regulatory substances and root activity. 

The activity levels of superoxide dismutase (SOD), peroxidase (POD) and catalase (CAT) were determined using assay kits for each respective enzyme (SOD-A001-3-2, POD-A084-3-1, CAT-A007-1-1, Nanjing Jiancheng Institute of Biological Engineering, Nanjing, China) according to the manufacturer’s instructions. They were calculated per gram of fresh biomass. 

The contents of reduced glutathione (GSH), ascorbic acid (AsA), H_2_O_2_, malondialdehyde (MDA), soluble sugar, soluble protein, Na^+^ and K^+^ were determined using GSH-A006-2-1, AsA-A009-1-1, H_2_O_2_-A064-1-1, MDA-A003-3-1, soluble sugar-A145-1-1, soluble protein-A045-2-2, Na^+^-C002-1-1 and K^+^-C001-2-1 kits (Nanjing Jiancheng Institute of Biological Engineering, Nanjing, China), respectively, according to the manufacturer’s instructions [68,69,70,71,72]. They were calculated per gram of fresh biomass.

The triphenyltetrazolium chloride (TTC) method was used to measure root activity [38]. The content of melatonin was determined using a TP1023 kit (Beijing Regen Biotechnology Co., Ltd., Beijing, China) according to the manufacturer’s instructions. They were both calculated per gram of fresh biomass.

##### Determination of Hormone Content

At 0, 5, 10 and 15 d after treatment initiation, 0.5 g samples were collected from three replicates under each treatment. After grinding each sample into a powder with liquid nitrogen, it was centrifuged with sample extraction solution, and the supernatant was collected, passed through a C-18 solid phase extraction column and then transferred to a centrifuge tube. The residual liquid in the centrifugal tube was blown dry with a nitrogen blower to remove methanol, and the volume was determined according to its fresh weight:diluent ratio (1:3). The contents of auxin (IAA), abscisic acid (ABA), gibberellin (GA) and brassinosteroid (BR) were determined by ELISA [73]. 

##### Ultrastructure Observations

Samples were collected 15 d after salt stress treatment. Small 2–5 mm sections from the root tips of seedlings grown under different treatments were sampled, fixed with 2.5% (*v/v*) glutaraldehyde and dehydrated with a graded ethanol series. The samples were permeated in a graded mixture of acetone and SPI-PON812 resin and embedded in pure resin. Samples were sectioned with an ultrathin slicing machine (Leica EM UC6: Leica, Wetzlar, Germany) into 70 nm thick sections, stained with uranyl acetate and lead citrate and observed with a transmission electron microscope (FEI Tecnai Spirit 120 kV; FEI, Hillsboro, OR, USA) [74]. 

##### Transcriptome Analysis

According to the cotton root morphology of different treatments, transcriptional and metabolic analyses were performed at 48 h (2 d) after treatment [75]. Roots of seedlings subjected to each treatment were quickly and repeatedly rinsed in distilled water three times to remove the surface nutrient solution, and afterwards, the surface moisture of roots was dried with absorbent paper. Then, 0.5 g samples were accurately weighed and frozen with liquid nitrogen. Three biological replicates were collected from each treatment. Transcriptome data were generated by Gene Denovo Biotechnology Co., Ltd. (Guangzhou, China). 

##### RNA Extraction, Library Construction and Sequencing

Total RNA was extracted using a TRIzol reagent kit (Invitrogen, Carlsbad, CA, USA). RNA integrity and DNA contamination were assessed by agarose gel electrophoresis, and RNA integrity was accurately detected using the Agilent 2100 BioAnalyzer platform (Agilent, Santa Clara, CA, USA). The library was sequenced on the Illumina Novaseq 6000 platform (Illumina, San Diego, CA, USA). To ensure data quality, clean reads were obtained by removing reads with a proportion of N base calls greater than 10%, all reads with A base calls and low-quality reads (i.e., reads with bases with a quality value score Q ≤ 20 accounting for more than 50% of the whole read). After removing the low mass readings, 543 M reads of clean data were obtained.

The mapped reads of each sample were assembled using StringTie v1.3.1 according to a reference-based approach. For each transcription region, a fragment per kilobase of transcript per million mapped reads (FPKM) value was calculated to quantify its expression abundance. According to the comparison results of HISAT2, Stringtie was used to reconstruct the transcripts, and RSEM was used to calculate the expression levels of all genes in each sample.

RNA differential expression analysis was performed by DESeq2 software between two different groups (and by the edgeR package in R for comparisons between two samples). The genes/transcripts determined based on a false discovery rate (FDR) < 0.05 and absolute fold change ≥2 were considered differentially expressed genes/transcripts.

Gene Ontology (GO) enrichment analysis was used to determine all GO terms that were significantly enriched among DEGs, and DEGs were filtered according to their biological functions. All DEGs were mapped to GO terms in the Gene Ontology database (http://www.geneontology.org/ (accessed on 19 August 2021)), and gene numbers were calculated for every term; significantly enriched GO terms among DEGs compared to the genome background were identified by the hypergeometric test. The Kyoto Encyclopedia of Genes and Genomes (KEGG) annotation includes not only genes but also metabolites, so differentially expressed metabolites were mapped to KEGG metabolic pathways for both pathway analysis and enrichment analysis. 

##### qRT-PCR

Cotton CDS data were obtained from the Cotton Functional Genomics Database (http://cottonfgd.org (accessed on 19 August 2021), and primers were designed using Primer BLAST software on the NCBI website and synthesized by Shanghai Shenggong Bioengineering Co., Ltd. (Shanghai, China) (see Appendix A for primer sequences).

##### Metabolic Analysis

The freeze-dried samples were crushed using a mixer mill (MM 400; Retsch, Haan, Germany) with zirconia beads for 1.5 min at 30 Hz. Then, 100 mg of the resulting powder was weighed and extracted overnight at 4 °C with 1.0 mL of 70% aqueous methanol containing 0.1 mg/L lidocaine as an internal standard. Following centrifugation at 10,000× *g* for 10 min, the supernatants were absorbed and filtered (SCAA-104, 0.22-μm pore size; ANPEL, Shanghai, China, www.anpel.com.cn/ (accessed on 19 August 2021) before LC–MS/MS analysis. Quality-control (QC) samples were obtained by mixing all samples to detect the reproducibility of the whole experiment. Metabolic analysis was also carried out at 48 h after treatment, and 3 biological replicates were taken for correlation analysis for each treatment.

The metabolite detection data acquisition instrument system mainly included an ultra-high performance liquid chromatograph (UHPLC) and tandem mass spectrometer; multivariate statistical analysis (O) was used to combine partial least squares–discriminant analysis (PLS-DA) with univariate statistical analysis. The compounds extracted were analyzed using an LC–ESI–MS/MS system (UPLC, Shim-pack UFLC SHIMADZU CBM20A; Shimadzu, Beijing, China; MS/MS Applied Biosystems 4500 QTRAP; Applied Biosystems, Beijing, China).

### 4.3. Statistical Analysis

Microsoft Excel 2010 (Microsoft Corp., Redmond, WA, USA) and Adobe Illustrator 2020 (Adobe, San Diego, CA, USA) were used for data summary statistic generation, collation, analysis and mapping. IBM SPSS Statistics 21.0 software (IBM Corp, Armonk, NY, USA) was used for univariate ANOVA analysis among treatment groups. Combined analysis of transcriptome and metabolome data was performed using R 4.1.0 software.

## Figures and Tables

**Figure 1 ijms-23-09456-f001:**
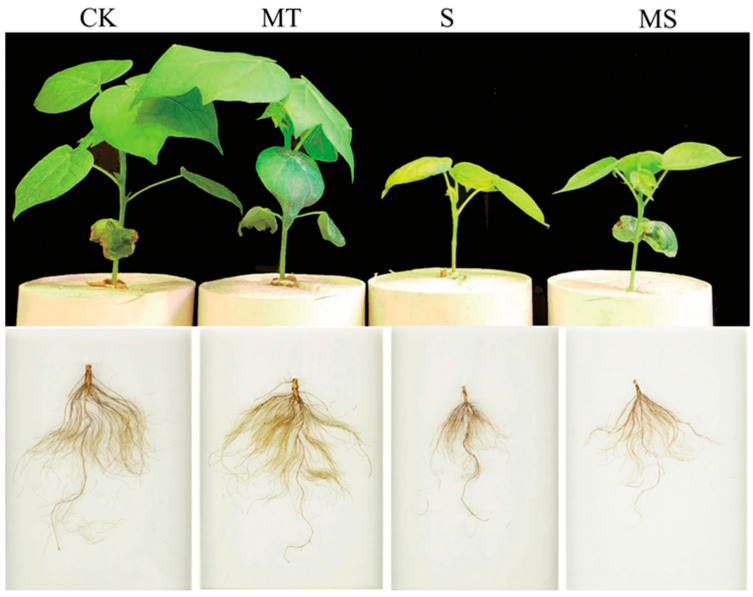
Dynamic changes in cotton plant shoots and roots under different treatments. The upper parts of cotton seedlings and their root systems after treatment with control conditions (CK), melatonin (MT), NaCl (S) and both melatonin and NaCl (MS) for 15 d.

**Figure 2 ijms-23-09456-f002:**
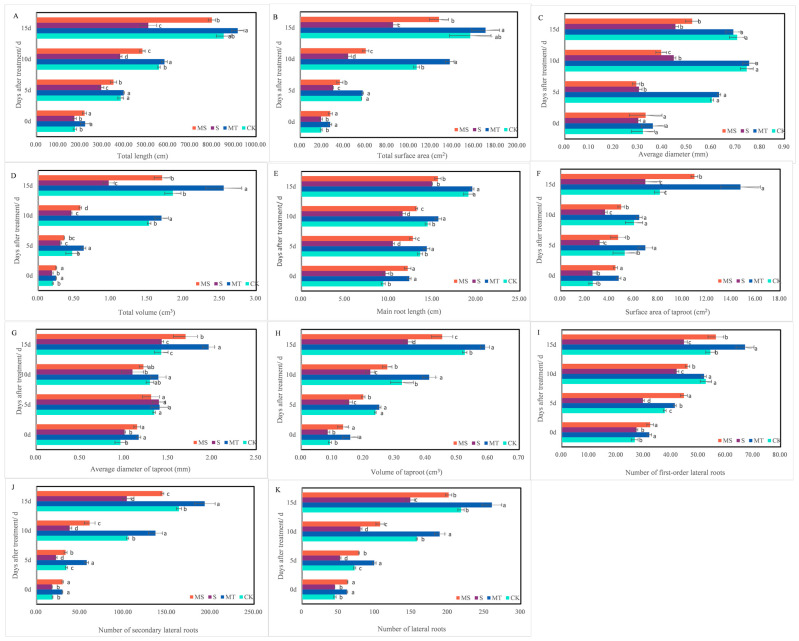
Effects of melatonin on cotton root morphology across treatment times. The upper parts of cotton seedlings and their root systems after treatment with control conditions (CK), melatonin (MT), NaCl (S) and both melatonin and NaCl (MS) for 15 d. (**A**) total length; (**B**) total surface area; (**C**) average diameter; (**D**) total volume; (**E**) main root length; (**F**) surface area of taproot; (**G**) average diameter of taproot; (**H**) volume of taproot; (**I**): number of first-order lateral roots; (**J**) number of secondary lateral roots; (**K**) number of lateral roots. Different lowercase letters indicate significant differences among treatments at the same timepoint (*p* < 0.05).

**Figure 3 ijms-23-09456-f003:**
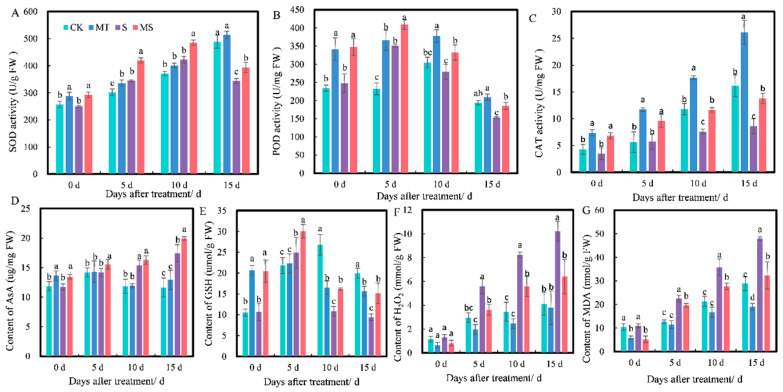
Effects of exogenous melatonin on (**A**) superoxide dismutase (SOD) activity, (**B**) peroxidase (POD) activity, (**C**) catalase (CAT) activity, (**D**) ascorbic acid (AsA) content, (**E**) reduced glutathione (GSH) content, (**F**) H_2_O_2_ content and (**G**) malonaldehyde (MDA) content of cotton seedling roots under salt stress. Different lowercase letters indicate significant differences among treatments at the same timepoint (*p* < 0.05). CK, control treatment; MT, melatonin treatment; S, NaCl treatment; MS, combined melatonin and NaCl treatment.

**Figure 4 ijms-23-09456-f004:**
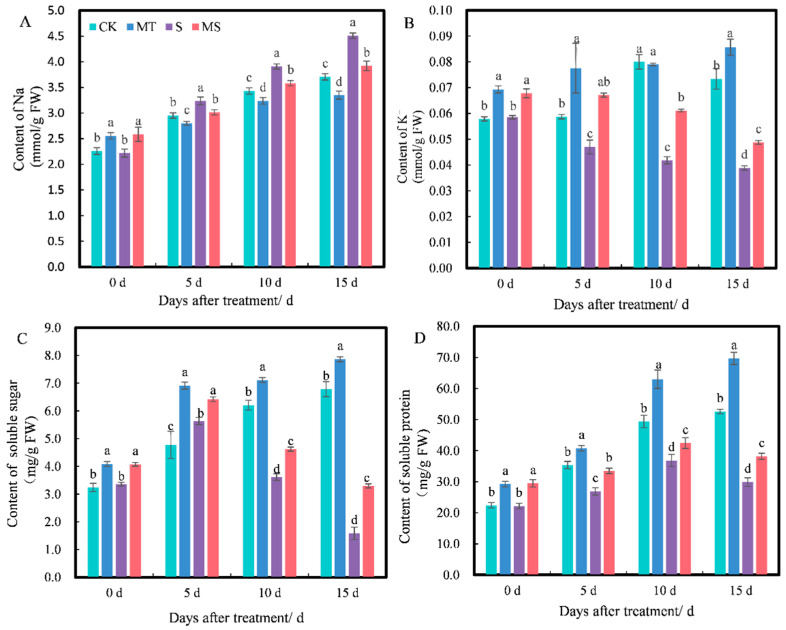
Effects of exogenous melatonin on Na^+^, K^+^, soluble sugar and soluble protein content of cotton seedling roots under salt stress: (**A**) Na^+^; (**B**) K^+^; (**C**) soluble sugar; (**D**) soluble protein. Different lowercase letters indicate significant differences among treatments at the same timepoint (*p* < 0.05). CK, control treatment; MT, melatonin treatment; S, NaCl treatment; MS, combined melatonin and NaCl treatment.

**Figure 5 ijms-23-09456-f005:**
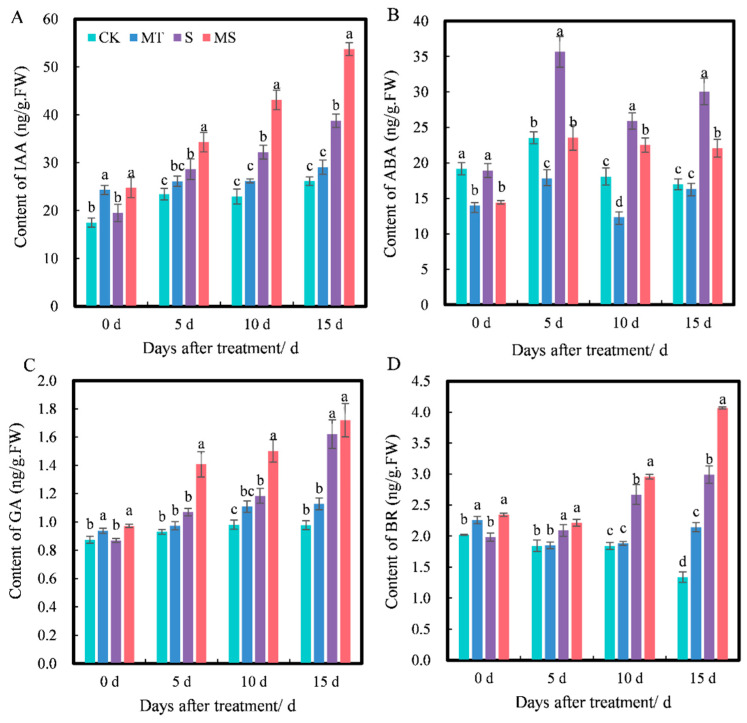
Effects of exogenous melatonin on (**A**) auxin (IAA), (**B**) abscisic acid (ABA), (**C**) gibberellin (GA) and (**D**) brassinosteroid (BR) contents of cotton seedling roots under salt stress. Different lowercase letters indicate significant differences among treatments at the same timepoint (*p* < 0.05). CK, control treatment; MT, melatonin treatment; S, NaCl treatment; MS, combined melatonin and NaCl treatment.

**Figure 6 ijms-23-09456-f006:**
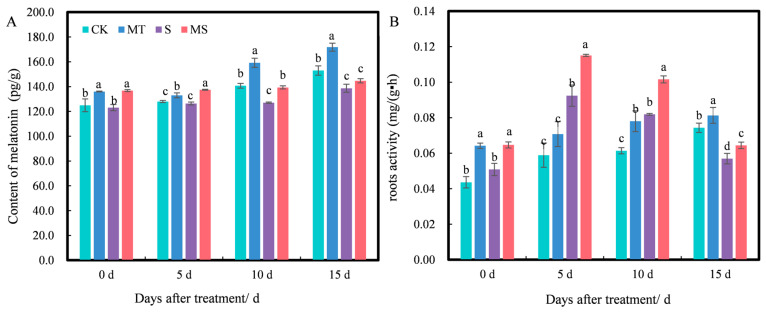
Effects of exogenous melatonin on (**A**) endogenous melatonin contents and (**B**) root activity of cotton seedling roots under salt stress. Different lowercase letters indicate significant differences among treatments at each timepoint (*p* < 0.05). CK, control treatment; MT, melatonin treatment; S, NaCl treatment; MS, combined melatonin and NaCl treatment.

**Figure 7 ijms-23-09456-f007:**
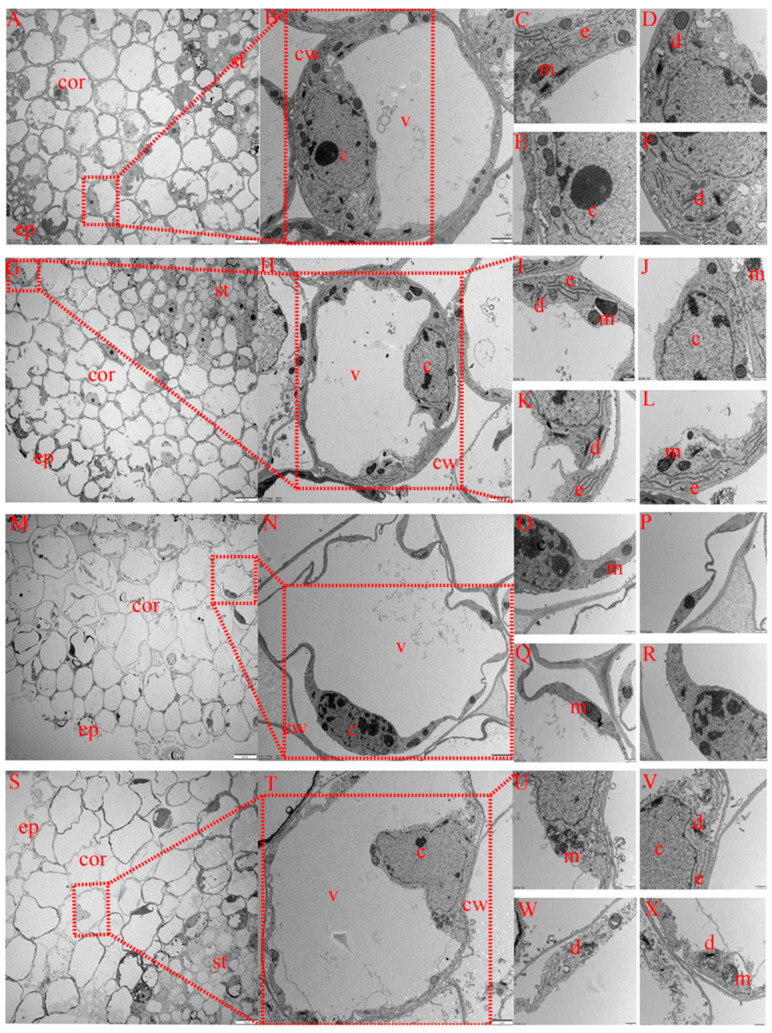
Effects of melatonin on root cortex cell ultrastructure of cotton seedlings under salt stress. (**A**–**F**) Transmission electron micrograph (TEM) of the cortex cell structure under control (CK) treatment; (**G**–**L**) TEM of the cortex cell structure under melatonin (MT) treatment; (**M**–**R**) TEM of the cortex cell structure under NaCl (S) treatment; (**S**–**X**) TEM of the cortex cell structure under the combined melatonin and NaCl (MS) treatment. ep, epidermis; cor, cortex; st, stele; m, mitochondrion; e, endoplasmic reticulum; d, dictyosome; c, cell nucleus; v, vacuole.

**Figure 8 ijms-23-09456-f008:**
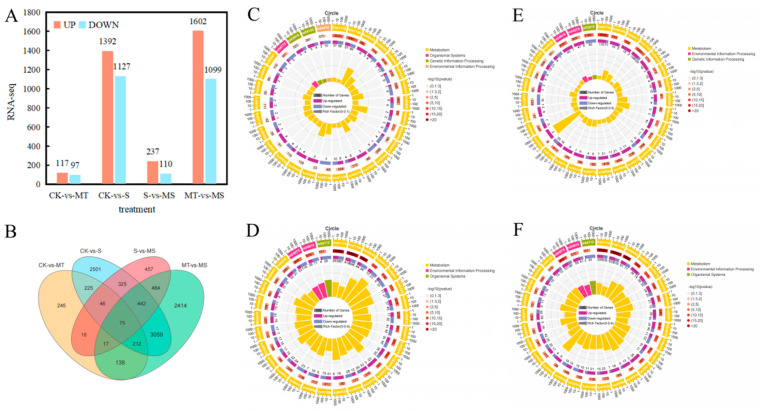
RNA-seq data analysis of roots in cotton seedlings under different treatments: (**A**) the number of differentially expressed genes (DEGs) in the CK vs. S, CK vs. MT, S vs. MS and MT vs. MS comparisons; (**B**) Venn diagram of DEGs; summary of Kyoto Encyclopedia of Genes and Genomes (KEGG) categories of DEGs in (**C**) CK vs. MT, (**D**) CK vs. S, (**E**) S vs. MS and (**F**) MT vs. MS comparisons. CK, control treatment; S, NaCl treatment; MT, melatonin treatment; MS, combined NaCl and melatonin treatment.

**Figure 9 ijms-23-09456-f009:**
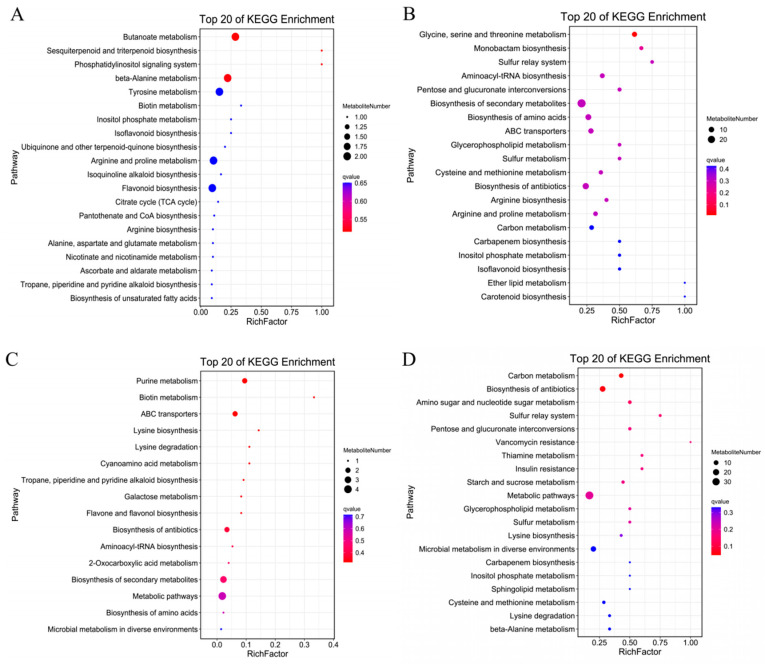
Differential metabolite enrichment bubble diagram of cotton seedling roots under (**A**) CK vs. MT, (**B**) CK vs. S, (**C**) S vs. MS and (**D**) MT vs. MS treatments. Note: KEGG, Kyoto Encyclopedia of Genes and Genomes; CK, control treatment; MT, melatonin treatment; S, NaCl treatment; MS, combined melatonin and NaCl treatment.

**Figure 10 ijms-23-09456-f010:**
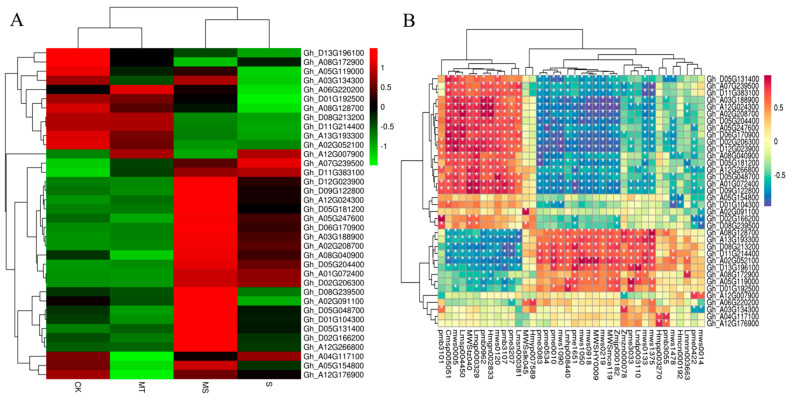
Correlation analysis of target gene expression levels in cotton seedling roots under control, salt, melatonin and combined salt and melatonin treatments. (**A**) Heat map of target genes under different treatments. (**B**) Correlation analysis diagram of target genes and metabolites.

**Figure 11 ijms-23-09456-f011:**
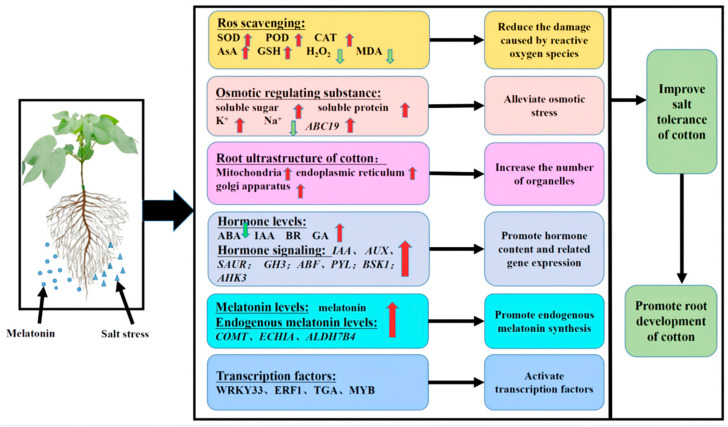
Exogenous melatonin regulates the physiological and molecular patterns of cotton seedling roots and seedlings. Note: SOD, superoxide dismutase; POD, peroxidase; CAT, catalase; GSH, the contents of reduced glutathione; AsA, ascorbic acid; MDA, malondialdehyde; IAA, auxin; ABA, abscisic acid; GA, gibberellin; BR, brassinosteroid; *Aux/IAA*, auxin protein gene; *SAUR*, auxin small RNA gene; *GH3*, auxin response gene; *AHK*, cytokinin receptor genes; *GID1*, GA receptor gene; *ABF*, ABA response element binding factor gene; *BZR*, Br-associated receptor kinase gene; *PYL*, abscisic acid receptor gene. Red and green arrows indicate increases and decreases, respectively.

**Table 1 ijms-23-09456-t001:** Experimental treatment with control conditions (CK), melatonin (MT), NaCl (S) and both melatonin and NaCl (MS).

Treatments	CK	MT	S	MS
For 24 h	0 μmol∙L^−1^ melatonin (distilled water)	10 μmol∙L^−1^ melatonin	0 μmol∙L^−1^ melatonin (distilled water)	10 μmol∙L^−1^ melatonin
Two-true-leaf stage	0 μmol∙L^−1^ NaCl	0 μmol∙L^−1^ NaCl	150 mmol∙L^−1^ NaCl	150 mmol∙L^−1^ NaCl

## Data Availability

Data supporting reported results of this study are available in the Appendix A of this article and can be obtained from the corresponding author.

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
