# Peer review of "Effects of Exogenous Melatonin on Root Physiology, Transcriptome and Metabolome of Cotton Seedlings under Salt Stress"

_ijms, 2022, doi:10.3390/ijms23169456_

Round 1
Reviewer 1 Report
Concerning the present manuscript, I have just one major suggestion, plus a couple of minor ones.
1) Authors should provide an explanation about why they presented the root morphology data in tabular form, which is quite counfounding, instead of in graphical form, much more simple to understand and evaluate
2) Please remove "and so on" from line 140: never use such generic expression in a scientific manuscript!
3) Please explain the meaning of the heading "Results and Analysis" appearing on line 232...What does it mean? These are simply "Results", in my opinion, because the methodological part has been already described in the M&M section...Isn't it?
All the above considering, I recommend further consideration of the present manuscript after major revision, adequately taking into account the points raised above.
Author Response
Dear Reviewers
On behalf of my co-authors, we thank you very much for giving us an opportunity to revise our manuscript again, we appreciate Reviewers very much for their positive and constructive comments and suggestions on our manuscript entitled “Effects of exogenous melatonin on root physiology, transcriptome and metabolome of cotton seedlings under salt stress”. Those comments are all valuable and very helpful for revising and improving our paper, as well as the important guiding significance to us researches. We have studied comments carefully and have made correction which we hope meet with approval. Revised portion are marked in red in the paper.
Please see the attachment

Reviewer 2 Report
Dear Authors
Manuscript entitled “Effects of exogenous melatonin on root physiology, transcriptome and metabolome of cotton seedlings under salt stress” investigated the the effects of exogenous melatonin on the root physiology at transcriptome and metabolome level. The study revealed that exogenous melatonin could mitigate the inhibition of salt stress on plant root development by regulating the reactive oxygen species scavenging system, ABC transporter synthesis, plant hormone signal transduction, endogenous melatonin gene expression and expression of the transcription factors MYB, TGA and WRKY33.The study is quite interesting in the area, although there are certain points to be answered, please find my suggestion here.
1- Abstract is very well written with the key findings of the study. Introduction is concise, focused and informative.
2- Materials and Methods section need more details. Determination of enzyme activity should include more details regarding sample preparations, only references are not enough for the readers.
3- Results, discussion are quite well presented and concluded nicely.
4- Major question is feasibility of Melatonin for exogenous application. It may be an expensive method due to high cost of extraction in pure form. Second, the melatonin application itself induced the melatonin synthesis in the plants? It may be suggested further to quantify the total melatonin before and after application, probably in next studies.
Thank you
Regards
Author Response
Dear Reviewer
On behalf of my co-authors, we thank you very much for giving us an opportunity to revise our manuscript again, we appreciate Reviewers very much for their positive and constructive comments and suggestions on our manuscript entitled “Effects of exogenous melatonin on root physiology, transcriptome and metabolome of cotton seedlings under salt stress”. Those comments are all valuable and very helpful for revising and improving our paper, as well as the important guiding significance to us researches. We have studied comments carefully and have made correction which we hope meet with approval.
Please see the attachment.

Reviewer 3 Report
Review on the manuscript “ Effects of exogenous melatonin on root physiology, transcriptome and metabolome of cotton seedlings under salt stress”
The authors conducted a study on the effect of exogenous melatonin on important cultivated plant cotton under salt stress. Morphological, biochemical, transcriptome and metabolome methods were used and a large amount of data was obtained. This study may be interesting for agricultural and plant selection specialists. However, there are many questions and comments to experiment design, methods and text of the manuscript.
1) In Abstract, line 19, the authors wrote that «…After subjecting melatonin-treated seeds to salt stress,…» and in Methods, lines 121-122 «…After wiping away their surface moisture, the seeds were soaked in 0 μmol∙L-1 or 10 μmol∙L-1 [32] melatonin solution for 24 h at 25°C in the dark…». Then 4 treatments are described (lines 129-132) : «…At the two true-leaf stage, seedlings were subjected to the following treatments: 0 μmol∙L-1 NaCl and 0 μmol∙L-1 melatonin (CK); 0 μmol∙L-1 NaCl and 10 μmol∙L-1 melatonin (MT); 150 mmol∙L 1 NaCl [33] and 0 μmol∙L-1 melatonin (S); 150 mmol∙L-1 NaCl and 10 μmol∙L-1 melatonin (MS).». However, nowhere is it indicated that these treatments were made on plants from two groups of seeds: untreated seeds and melatonin-treated seeds. In Results the data for only 4 treatments variants (CK, MT, S, MS) are shown. This indicates that these are the results only for treatments of one group of seeds.
Question: what seeds were used for these 4 treatments: untreated seeds (0 μmol∙L-1 melatonin solution) or melatonin-treated seeds (10 μmol∙L-1 melatonin solution)? If untreated seeds were used, why do the authors write about melatonin-treated seeds? If melatonin-treated seeds were used, then the authors should add the results of the same 4 treatments for untreated seeds, which are the control in this case.
2) Line 138 - At 0, 5, 10, and 15 d after treatment…. What does 0 d after treatment mean? Is it before treatment or how many minutes-hours after treatment? If it is before treatment - how can the authors explain the differences in almost all studied biochemical parameters between treated plants at 0 d after treatment (Figures 2-5)?
3) Why was transcriptome analysis done at 48 h after the start of treatment (line 168) and not 0, 5, 10, and 15 d after treatment? Why do the authors think that it is possible to compare the transcriptome analysis results at 48 h (2 d) after treatment and the biochemical analysis results at 5, 10, and 15 d after treatment? Why were these analysis not done at the same time?
4) At what day after treatment there was done metabolic analysis and how many biological replicates were taken for analysis for each treatment? There is no such information in the text.
5) How many biological replicates were taken for correlation analysis for each treatment? There is no such information in the text. Correlation analysis requires a large number of biological replicates, so the results are questionable.
6) Sections 2.2.2.5 (qRT-PCR) and 3.7.2. Gene express analysis was done for “… To verify the accuracy of differentially expressed genes identified by RNA-seq…” (line 387), which is essentially internal control and does not provide new information, therefore these sections, including Fig. 8 and tab. 1 and 3 (which are better combined) should be moved to Supplemental material.
7) On Figures 2-5 the significance of differences between 4 treatments (CK, MT, S, MS) at each 0, 5, 10, and 15 d after treatment are shown. Why don't the authors show the significance of differences for each treatment among all days of treatment? Such analysis may show interesting results.
8) Figure 5 – root activity - in the methods there is no information about the calculations of the root activity. The authors should add it.
9) Figures 2-3. Contents of SOD, POD, CAT, AsA, GSH, H2O2, MDA, Na+, K+ are calculated per gram of what? Fresh or dry biomass?
10) Fig. 7 - the authors should add a list of genes that are given in Figure 7 and described in the text (3.7.1. section).
11) Fig. 11 – trends for AsA, GA do not match the previous figures.
12) 4.6. Regulation mode of exogenous melatonin on root growth of cotton under salt stress
Lines 698-699 – “…Additionally, exogenous melatonin can inhibit the accumulation of H2O2 and MDA…” This is not correct; rather, the decrease in H2O2 and MDA content is a consequence of the higher activity of antioxidant enzymes, as the authors write on lines 697-698.
The authors should be more careful with the term "inhibit" (line 701 - inhibit the accumulations of Na+ content… line 703 - The melatonin treatment also inhibited the accumulations of ABA…) - as it requires evidence at the level of specific mechanisms, which the authors do not provide.
Section Conclusions consists of very general proposals. Authors should either specify the conclusions and emphasize novelty, or remove them. Section 4.6. “Regulation mode of exogenous melatonin on root growth of cotton under salt stress” in fact is generalizing and it is enough.

Author Response

(The authors gave the same response as above.)

Round 2
Reviewer 1 Report
After examining the revised version of the present manuscript, I found a couple of points which, in my opinion, still deserve adequate consideration on the part of the Authors:
1) It is not clear to me why in the original submission the material identified as “supplementary” (Tables S1, S2 and so forth) were placed in the main text, whereas, in the present revised version, such supplementary material has been removed from the main text. Please explain.
2) In any case, it is inappropriate, and even a bit weird, in my opinion, to exile the results concerning root morphology (i.e. Table S2) into the supplementary material, simply because the “core business” of the present manuscript is heavily based on root structure and function. So that I would strongly suggest to place the results in Table S2 in the main text.
3) as for the form in which Table S2 results should be presented, I have difficulties in accepting the justification put forward by the Authors for their preference for tabular form. Indeed, and as far as I can imagine, IJMS, being an on line-only Journal, should have no problem of space limitation. Furthermore, by using the same graphical approach used for the other figures of the present manuscript, it is quite probable that the eleven parameters measured here for root morphology would fill exactly the same space taken by the rather cumbersome Table S2, i.e. just one page (or even less), but certainly with a much clearer and attractive appearance. So that, I would strongly suggest to convert Table S2 into a graphical form.
All the above considering, I recommend further major revision of the present manuscript..
Author Response
-
Dear Reviewers
On behalf of my co-authors, we thank you very much for giving us an opportunity to revise our manuscript again, we appreciate Reviewers very much for their positive and constructive comments and suggestions on our manuscript entitled “Effects of exogenous melatonin on root physiology, transcriptome and metabolome of cotton seedlings under salt stress”. Those comments are all valuable and very helpful for revising and improving our paper, as well as the important guiding significance to us researches. We have studied comments carefully and have made correction which we hope meet with approval. The main corrections in the paper and the responds to the Reviewer’s comments are as flowing:
- It is not clear to me why in the original submission the material identified as “supplementary” (Tables S1, S2 and so forth) were placed in the main text, whereas, in the present revised version, such supplementary material has been removed from the main text. Please explain.
Response: Thank you for the very helpful suggestion. We have noted that the editor has responded to your question and thank you again for your suggestion
- In any case, it is inappropriate, and even a bit weird, in my opinion, to exile the results concerning root morphology (i.e. Table S2) into the supplementary material, simply because the “core business” of the present manuscript is heavily based on root structure and function. So that I would strongly suggest to place the results in Table S2 in the main text.
Response: Thank you for the very helpful suggestion. We have placed the results in Table S2(changed to Figure 2) in the main text.
3) as for the form in which Table S2 results should be presented, I have difficulties in accepting the justification put forward by the Authors for their preference for tabular form. Indeed, and as far as I can imagine, IJMS, being an on line-only Journal, should have no problem of space limitation. Furthermore, by using the same graphical approach used for the other figures of the present manuscript, it is quite probable that the eleven parameters measured here for root morphology would fill exactly the same space taken by the rather cumbersome Table S2, i.e. just one page (or even less), but certainly with a much clearer and attractive appearance. So that, I would strongly suggest to convert Table S2 into a graphical form.
Response: Thank you for the very helpful suggestion. We have changed the Table S2 into Figure 2 according to the Reviewer’s comments.
Reviewer 3 Report
Please see comments to revised manuscript in attached file

Author Response
Dear Reviewers
On behalf of my co-authors, we thank you very much for giving us an opportunity to revise our manuscript again, we appreciate Reviewers very much for their positive and constructive comments and suggestions on our manuscript entitled “Effects of exogenous melatonin on root physiology, transcriptome and metabolome of cotton seedlings under salt stress”. Those comments are all valuable and very helpful for revising and improving our paper, as well as the important guiding significance to us researches. We have studied comments carefully and have made correction which we hope meet with approval.
Please see the attachment.

Round 3
Reviewer 1 Report
In he present second revision of the present manuscript, the suggestions offered on the preceding version have been adequately met by the Authors..
Therefore, I recommend acceptance of the manuscript in its present form